# Antimicrobial Susceptibility from a One Health Perspective Regarding Porcine *Escherichia coli* from Bavaria, Germany

**DOI:** 10.3390/antibiotics12091424

**Published:** 2023-09-08

**Authors:** Julia Ade, Julia M. Riehm, Julia Stadler, Corinna Klose, Yury Zablotski, Mathias Ritzmann, Dolf Kümmerlen

**Affiliations:** 1Clinic for Swine, Centre for Clinical Veterinary Medicine, Ludwig-Maximilians-Universität München Sonnenstrasse 16, 85764 Oberschleissheim, Germany; julia.ade@outlook.com (J.A.); j.stadler@med.vetmed.uni-muenchen.de (J.S.);; 2Bavarian Health and Food Safety Authority, Veterinaerstrasse 2, 85764 Oberschleissheim, Germanycorinna.klose@lgl.bayern.de (C.K.); 3Clinic for Ruminants with Ambulatory and Herd Health Services, Centre for Clinical Veterinary Medicine, Ludwig-Maximilians-Universität München, Sonnenstrasse 16, 85764 Oberschleissheim, Germany; 4Division of Swine Medicine, Department for Farm Animals, Vetsuisse Faculty, University of Zurich, Winterthurerstrasse 260, 8057 Zurich, Switzerland

**Keywords:** *Escherichia coli*, antimicrobial resistance, epidemiological cut-off, pigs

## Abstract

Antimicrobial resistance is one of the most crucial One Health topics worldwide. Consequently, various national and international surveillance programs collect data and report trends regularly. Ceftiofur, colistin and enrofloxacin belong to the most important and critical class of anti-infective medications in both human and veterinary medicine. In the present study, antimicrobial resistance was analyzed using the epidemiological cut-off (ECOFF) value on 6569 *Escherichia coli* isolated from pigs in Bavaria, Germany, during five years, from 2016 to 2020. The statistically relevant results regarding antimicrobial resistance revealed a decrease for colistin, an increase for enrofloxacin, and a constant level for ceftiofur. In Germany, the usage of all three antimicrobial substances in livestock has fallen by 43.6% for polypeptides, 59.0% for fluoroquinolones and 57.8% for the 3rd + 4th generation cephalosporines during this time. Despite the decline in antimicrobial usage, a reduction regarding antimicrobial resistance was solely observed for colistin. This finding illustrates that in addition to the restriction of pharmaceutical consumption, further measures should be considered. Improved biosecurity concepts, a reduction in crowding, and controlled animal movements on farms may play a key role in finally containing the resistance mechanisms of bacteria in farm animals.

## 1. Introduction

Antimicrobial resistance in bacteria isolated from humans, animals and the environment is continuously rising. This problem and global issue in human and veterinary medicine is a major concern of the “One Health” initiative [1,2,3,4]. Farm animal husbandry was repeatedly criticized for contributing to the development and spread of antimicrobial resistance using antimicrobials in animal production [5,6,7,8]. In order to reduce this increase, it is important to further control the use of antimicrobials in livestock farming [9,10]. The restriction of certain classes of substances is also a recurring topic of discussion. Both the World Health Organization (WHO) for human health and the World Organization for Animal Health (WOAH) define individual antimicrobial agents and antimicrobial classes as “critical substances” and their use is under ongoing scrutiny [9,11]. Consequently, the sale and usage of these antimicrobials has decreased in recent years in Europe. Among those critical substances are enrofloxacin, ceftiofur and colistin [12].

Enrofloxacin belongs to the quinolone group and is used as a broad-spectrum antimicrobial substance due to its bactericidal effects [13,14]. Ceftiofur is a third-generation cephalosporine with a long-lasting efficacy and is traditionally used as an injectable [15,16]. Colistin is a polymyxin antimicrobial substance changing the structure of the outer cell membrane of Gram-negative bacteria, leading to leakage and consequently to cell-death [17]. Resistance may derive from chromosomal mutations as well as from plasmid-transmitted resistance genes [18,19]. Colistin is used as a last-choice therapy option in human medicine for the treatment of infections with multidrug-resistant Gram-negative bacteria. In livestock, colistin was widely used, especially for enteric infections, including prophylactic use in pigs [20,21,22,23,24,25,26]. The commensal Gram-negative bacterium *Escherichia coli* (*E. coli*) is associated with the gastrointestinal tract. It is a widely distributed bacterium in humans and animals. Apart from this role, *E. coli* may turn into an opportunistic pathogen and it is known to harbor and further transmit genetic elements of antimicrobial resistance on various routes [27,28,29]. Thus, *E. coli* is also used as an indicator bacterium for antimicrobial resistance surveillance programs, e.g., in surveillance programs run by the European Centre for Disease Prevention and Control (ECDC), the European Food Safety Authority (EFSA) and in national programs like GERM-Vet in Germany [27,30,31].

The characterization of bacteria in terms of their susceptibility towards antimicrobial substances depends on clinical breakpoints. These were defined considering the test group, the body site and the antimicrobial agent [32,33,34]. The breakpoints classify the investigated isolate as “sensitive”, “intermediate” or “resistant” against the tested agent in order to forecast the therapeutic success of the substance. However, clinical breakpoints depend on the host species as well as on the target tissue [32]. Therefore, an extrapolation on susceptibility in a disease event for humans or another animal species is limited. Further, for some antimicrobial substances, clinical breakpoints are not available, as, for example, for colistin [32]. As a further possibility to compare antimicrobial efficacy for bacterial species, irrespective of the individual breakpoints, the so-called epidemiological cutoffs (ECOFFS) may be applied, as defined by the European Committee on Antimicrobial Susceptibility Testing (EUCAST). Based on phenotypical characteristics in the form of minimal inhibitory concentration (MIC) distribution patterns, germs exhibiting MICs below the ECOFF are defined as “wild-type” (WT) and germs above the ECOFF are defined “non-wild-type” (NWT). As previously defined, NWT isolates predict the occurrence of acquired resistance mechanisms from the determined MIC, but do not consider the isolation origin, such as the host organism or the affected tissue [33,34,35].

In Germany, the global and European requirements for antimicrobial resistance monitoring in human and veterinary medicine are mainly implemented through the “German Antibiotic Resistance Strategy” (DART). DART provides action plans, surveillance and monitoring systems, which are based on random sample surveys. The monitoring of veterinary antimicrobial resistance includes a program for commensal pathogens as well as a separate program for pathogens known as GERM-Vet. GERM-Vet yearly reports present antimicrobial resistance data for different livestock species including pigs, mainly based on clinical breakpoints, or, if no clinical breakpoint is available, based on observed MIC_90_ values [36]. These data are essential to continuously monitor and evaluate the situation of antimicrobial resistance originating from livestock. As an additional approach, the present study aimed to analyze the antimicrobial resistance of *E. coli* isolates from diseased pigs in Bavaria, Germany, with a broader focus. For this purpose, the three “critical class” antimicrobial substances, enrofloxacin, ceftiofur and colistin, were chosen and antimicrobial resistance was determined using the ECOFF classification. Revealing trends in antimicrobial resistances during a certain period enables an estimation of whether current treatment recommendations may be wisely continued.

## 2. Results

### 2.1. Isolates

In the present study, a total number of 6569 *E. coli* were isolated from diseased or deceased pigs in Bavaria, Germany. Out of these, 995 strains were isolated from 2016, 516 from 2017, 1874 from 2018, 1718 from 2019 and 1466 from 2020. Raw data are presented in Appendix A. 

### 2.2. Colistin

For colistin, MIC values between <0.5 µg/mL and >2 µg/mL have been observed in the 6569 sampled *E. coli* isolates. The EUCAST epidemiological cut-off (ECOFF) for *E. coli* and colistin is 2 µg/mL [35]. Therefore, all isolates with an MIC >2 µg/mL were classified as NWT isolates. Table 1 shows the distributions of all MIC values for colistin for the sampled years 2016–2020. The grey shaded column presents the proportion of isolates categorized as NWT. Regarding the overall percentages of NWT isolates between 2016 and 2020, a constant decrease in the number was observed with the highest number of NWT isolates in 2016 (6.63%) and the lowest number in 2020 (3.01%).

Statistical evaluation of the distribution of NWT isolates among the sampled *E. coli*, using a linear mixed effects logistic regression model, determined the following probabilities for the occurrence of NWT isolates: 4.49% for 2016, 4.1% for 2017, 2.78% for 2018, and 2.24% and 2.28% for 2019 and 2020, respectively. Figure 1 shows the probabilities for the occurrence of NWT isolates from 2016 to 2020 including the corresponding confidence intervals.

Further statistical evaluations with the linear mixed effect logistic regression model included investigations between the years 2016 and 2020 to point out a possible development of NWT isolates. This included the calculation of odds ratio, as well as change and *p*-value. The herein obtained values are listed in Table 2 below. Taken together, the results show a statistically significant decrease in the occurrence of NWT isolates when comparing 2020 to 2016.

### 2.3. Enrofloxacin

The overall distribution of determined MIC for enrofloxacin between 2016 and 2020 is shown in Table 3. Grey shaded columns highlight the isolates classified as NWT according to the ECOFF for enrofloxacin. NWT isolates have varied over the years. The overall number of NWT isolates was 18.20% in 2016, 16.50% in 2017, 29.01% in 2018 and 27.51% and 27.37% in 2019 and 2020, respectively.

Calculated year probabilities for the occurrence of NWT isolates for enrofloxacin decreased between 2016 and 2017 from 18.1% to 16.52% and increased to 24.52% in 2018. In 2019 and 2020, they were 23.05% and 24.03%, respectively. Probabilities for the occurrence of NWT isolates from 2016 to 2020 and the corresponding confidence intervals are displayed in Figure 2.

The trend analysis considering the five-year period between 2016 and 2020 revealed the following odds ratio, change and *p*-value for the occurrence of NWT enrofloxacin isolates of *E. coli*, as displayed in Table 4. According to this, a statistically significant increase in the occurrence of NWT isolates was shown.

### 2.4. Ceftiofur

The observed MIC values for ceftiofur are shown in Table 5. The relative number of NWT isolates was below 10% in 2016 (8.76%), 2017 (5.87%) and 2020 (9.62%) and exceeded 10% in 2018 (11.22%) and 2019 (10.85%). All values are represented in Table 5.

Calculated year probabilities for the occurrence of NWT isolates for ceftiofur decreased from 7.59% to 5.1% from 2016 to 2017. In 2018 and 2019, they were at 8.63% and 8.51% and slightly decreased again to 7.33% in 2020. Probabilities for the occurrence of NWT isolates from 2016 to 2020 and the corresponding confidence intervals are illustrated in Figure 3.

Table 6 shows the odds ratio, change and *p*-value for the occurrence NWT ceftiofur isolates of *E. coli* between 2016 and 2020. The shown changes are not significant.

### 2.5. Co-Resistances

Bacterial resistance against all three substances was observed in a total of thirty *E. coli* isolates, wherein four of those isolates were obtained in 2016, one isolate in 2017, twelve isolates in 2018, nine isolates in 2019 and four isolates were obtained in 2020, respectively. Simultaneous resistance against ceftiofur and enrofloxacin was observed in a total of 337 *E. coli* isolates, wherein 36 of them are dated to 2016, 15 to 2017, 110 to 2018, 99 to 2019 and 77 to 2020. Co-resistance between ceftiofur and colistin was observed in a total of fifteen *E. coli* isolates, six of them from 2016, two from 2017, four from two from 2018 and 2020 each and one from 2019. Enrofloxacin and colistin co-resistance was observed in a total of 55 isolates, wherein 8 originated from the year 2016, 6 from 2017, 18 from 2018, 13 from 2019 and 10 from 2020.

## 3. Discussion

Monitoring and surveillance programs for antimicrobial susceptibility are essential for assessing the continuous evolution of antimicrobial resistance on national and global scales. One example is the German program for veterinary pathogens (GERM-Vet) that observes data on various randomized samples and follows clinical breakpoints [36]. As an addition to the results of this monitoring program, the present study aimed to evaluate the evolution of porcine NWT *E. coli* isolates for critically important antibiotics (i.e., colistin, enrofloxacin, ceftiofur) in a defined epidemiological scenario over a five-year period. The ECOFF breakpoints were chosen as they offer the possibility to evaluate antimicrobial resistances under a “One Health” perspective, while CLSI breakpoints better predict therapeutic success in the target species [32]. The selection of the investigated antimicrobials was based on their importance in both human and veterinary medicine. This importance is displayed by the categorization of all three substances as “highest priority critically important antimicrobials” according to the WHO Advisory Group on Integrated Surveillance of Antimicrobial Resistance (AGISAR) [11], as well as for ceftiofur and enrofloxacin in “Veterinary critically important antimicrobial agents” or colistin as “Veterinary highly important antimicrobial agents” according to the OIE List of Antimicrobial Agents of Veterinary Importance [37]. Further, all three substances are classified in Antimicrobial Advice ad hoc Expert Group, AMEG, category B, meaning that these substances are critically important in human medicine and thus their use should be restricted in animals [38].

Various genetic elements that code for mobilized colistin resistance (mcr) were found on transferable plasmids. As a reaction to the first publication of this finding in 2015, profound restrictions were imposed on the use of colistin in animals and this was adopted in many countries worldwide [39,40]. Consequently, the sales and use of colistin were significantly reduced [12,40,41]. Regarding our study period from 2016 to 2020, the sales of polymyxins in Germany declined by 7.59% from a total of 7.9 mg/population correction unit (PCU) to 7.3 mg/PCU [12]. Also, the actual usage of polymyxin antimicrobials in livestock decreased by 43.6% in Germany between 2014 and 2021 [42]. In the present study, a relative quantity of 3.01% to 6.63% of the isolates revealed a MIC value for colistin higher than 2 µg/mL. This criterion classifies the isolate as a “non-wild-type” according to the EUCAST ECOFF definition. Furthermore, the highest amount of NWT isolates was observed in 2016 (6.63% NWT *E. coli* isolates) and the lowest amount in 2020 (3.01% NWT *E. coli* isolates). This positive trend represents a constant decrease in the occurrence of NWT isolates. Statistical evaluation also showed a significant decrease in the probability of the occurrence of NWT isolates among all samples *E. coli* for colistin between 2016 and 2020 (Table 4). These findings are in line with other data published for Germany as data from the German monitoring system GERM-Vet show a decrease in MIC_90_ values for porcine pathogenic *E. coli* isolates during the same time [36]. Further, a comparable study from north-western Germany also showed a decrease in colistin resistance among porcine *E. coli* between 2015 and 2017 [43]. In other European countries, similar results have been published. In Spain, a reduction in colistin-resistant *E. coli* isolates between 2015 and 2017 [44], as well as a significant decrease in the detection of the mcr-1 gene in *E. coli* isolates in 2017 and 2021 were shown [45]. In France, a recent study showed a decrease in colistin resistance in porcine *E. coli* isolates between 2011 and 2018 [46]. Further, the EU summary report noted a statistically significant decrease in colistin resistance among commensal porcine *E. coli* isolates which were obtained from the caecum during slaughter [47]. This significant decrease in colistin resistance following a reduction in colistin use was explained in the past. The maintenance of colistin resistance often demands high fitness costs in bacteria, meaning that resistant genotypes do not last for long in the absence of the respective antimicrobial agent [41].

Similar to colistin, the use of enrofloxacin has been restricted in recent years. The number of sales for fluoroquinolones in Germany dropped between 2016 and 2020, from 24% from 1.0 mg/PCU to 0.8 mg/PCU [12]. Consequently, the usage in food-producing animals in Germany decreased by 59.0% between 2014 and 2021 [42]. The MIC values for enrofloxacin in the *E. coli* isolates in the present study revealed a relative quantity of 16.50% to 29.01% isolates with a MIC value higher than 0.25 µg/mL, which then classifies the isolate as a “non-wild-type” according to the EUCAST ECOFF system. A statistical analysis revealed a significant increase in the probability of the occurrence of enrofloxacin NWT isolates when comparing the data from 2016 to those of 2020. Similarly, Moenighoff and co-workers stated a significant increase in the enrofloxacin resistance in porcine *E. coli* isolates between 2006 and 2017 in north-western Germany [43], and also the final reports of the German monitoring show an increase in published MIC_90_ values for enrofloxacin between 2016 and 2020 [42]. Reasons for the herein observed negative correlation between the use of enrofloxacin and the evolution of the resistance trend could result from different microbiological characteristics. Fluoroquinolones are known to have a high degree of stability in the environment [48]. Further, fluoroquinolone resistance genes and resistance genes for other antimicrobials such as ß-lactams and tetracyclines can be found on the same plasmids [49,50,51,52]. Especially as tetracyclines and penicillins are the most frequently used antimicrobial classes in pig production [53], this could lead to a co-selection between fluoroquinolone resistance genes and resistance genes for other antimicrobials such as ß-lactamases. Therefore, enrofloxacin resistance could further spread without any exposure to the substance itself. Consequently, this can lead to a formation of further resistant bacteria in the stable environment, a phenomenon that has been described by other authors before [54,55]. Environmental spread, inter-/intraspecies spread, trade, export, etc., may lead to a broad distribution of resistant bacteria in Bavaria, Germany, and neighboring countries, and can cause a delay in the decrease in enrofloxacin resistance rates. However, it should also be mentioned that the usage of tetracyclines and penicillins was also decreased in the livestock sector by 56.3% and 37.2% between 2014 and 2021 [42].

Ceftiofur sales declined by 60.53% from 0.38 mg/PCU in 2016 to 0.15 mg/PCU in 2020 [12]. However, evolution trends determined by statistical analyses did not show a significant change in ceftiofur resistance when comparing isolates from 2016 to those from 2020 in the present study [56]. The lack of improvement regarding the susceptibility might be based on the bactericidal mode of action and may need more time to become visible [57]. It is important to know that although ceftiofur is used only in veterinary medicine, a structurally similar drug, ceftriaxone, is extensively used in human medicine. Both substances operate using the same mechanism [56]. Another plausible explanation for the nonexistent correlation to the massive reduction in ceftiofur sales could also be caused by the co-selection of ceftiofur resistance genes and other resistance genes, as described earlier for enrofloxacin. Further future effects on decreased usage in veterinary medicine regarding polypeptides (represented by colistin in the present study), fluoroquinolones (enrofloxacin), and 3rd + 4th generation cephalosporines (ceftiofur) will probably be more visible in the coming years.

The overall and worldwide increase in antimicrobial resistance recorded in human and veterinary medicine has led to antimicrobial stewardship programs in human medicine and antimicrobial monitoring programs, and a reduction in chemotherapeutics in agricultural animal production [58]. In human medicine, these programs are successful if implemented [58]. Regarding veterinary medicine, a reduction in resistance rates was observed in some cases. However, our study shows that a reduction in the antimicrobial usage of a certain antimicrobial class alone does not necessarily lead to a successful reduction in antimicrobial resistance against this antimicrobial. In addition, the substantial antimicrobial reduction cannot be continued in many countries at this rate. Animal health will be compromised if the therapeutic treatment of animals is further reduced. For this reason, further studies on the specific routes of transmission of antimicrobial resistance against different antimicrobial classes from livestock to humans need to be investigated and measures need to be developed on the herds to prevent transmission. Due to the complexity of the genesis, maintenance and spread of antimicrobial resistance, more factors need to be involved in future concepts for antimicrobial resistance reduction. Among other things, this can include internal hygiene and biosecurity measures. As an example, the occurrence and therefore also the potential spread of resistant bacteria were demonstrated on anesthetic masks for piglets [59]. Further, animal transport might also be considered as it is well known that this can contribute to the spread of germs [60]. While the present study indicates the occurrence of antimicrobial resistance from a total number of porcine *E. coli* isolates, future studies also need to evaluate different production groups (e.g., organic vs. non-organic production) and age groups of pigs. As already requested by other authors, a closer collaboration with human medicine is also required [4].

## 4. Materials and Methods

### 4.1. Study Design and Bacterial Isolates

In the present study, *E. coli* were isolated from diseased or deceased pigs with a preliminary diagnosis of various health problems affecting individual animals as well as stocks at the Bavarian Health and Food Safety Authority in Bavaria, Germany, during January 2016 and December 2020. Subsequent antimicrobial susceptibility testing was carried out and MIC values were determined for all *E. coli* isolates. The respective sampling area is shown in Figure 4.

*E. coli* was confirmed at a species level by positive fluorescence on ECD agar (Merck Millipore, Burlington, MA, USA) and MALDI-TOF mass spectrometry (Bruker, Bremen, Germany).

### 4.2. Susceptibility Testing and MIC Determination

Antimicrobial susceptibility testing was carried out by applying the microbroth dilution method and using the Micronaut-S, Grosstiere 4 system (Merlin, Bruker, Bornheim, Germany) according to the manufacturer’s instructions. The MICs of all *E. coli* isolates were determined for ceftiofur (class of cephalosporin III), colistin (polymyxin antimicrobials) and enrofloxacin (class of fluoroquinolones). The antimicrobial agents were chosen as representatives for critically important antimicrobials in human medicine according to the WHO definition and that of veterinary medicine according to the WOAH (i.e., polymyxins, cephalosporines III and IV, fluoroquinolones and macrolides).

### 4.3. Determination of Antimicrobial Resistance

EUCAST epidemiological cut-off values (ECOFFs) were assigned to the observed MIC values of all *E. coli* isolates for the listed antimicrobial agents, here being colistin, enrofloxacin and ceftiofur. Based on the ECOFF, bacteria were either categorized as “wild-type” (WT) or as “non-wild-type” (NWT) isolates. NWT isolates are suspected to have acquired resistance mechanisms and can be considered as microbiologically resistant isolates. ECOFFs were retrieved from the MIC EUCAST website in August 2023 [35]. ECOFFs for the herein investigated antimicrobials and *E. coli* are as follows: colistin 2 µg/mL; enrofloxacin 0.125 µg/mL; ceftiofur 1 µg/mL. The number of NWT isolates was determined for each year and compared with the other years using Microsoft^®^ Excel (2016). Co-resistances in pathogens were observed by regarding the determined ECOFFs for all three antimicrobials per isolate. Raw data are presented in Appendix A. 

### 4.4. Statistic Evaluation

All analyses were conducted using R Statistical software (R version 4.03., 2020; RStudio desktop version 1.4.1103, 2021). Logistic mixed effects models were used to explore the difference in antimicrobial susceptibility defined via ECOFFs among the years separately for *E. coli* and the antimicrobial substances ceftiofur, colistin and enrofloxacin. Contrasts (differences) between particular years were assessed after the model fit via the estimated least-squares marginal means (“emmeans” R package), with the Benjamini and Hochberg *p*-value correction for multiple comparisons [61]. An error level of 0.05 was used to declare statistical significance, while *α* < 0.1 was used for associations trending for statistical significance.

## 5. Conclusions

The present study on antimicrobial resistances in *E. coli* isolated from pigs, using epidemiological cut-offs for assessment, revealed a decrease for colistin, an increase for enrofloxacin, and a constant level for ceftiofur between the years 2016 and 2020. As the antimicrobial usage of all three substances in livestock decreased by at least one third during this time in Germany, further attempts should be made to control this important “One Health” problem of worldwide concern.

## Figures and Tables

**Figure 1 antibiotics-12-01424-f001:**
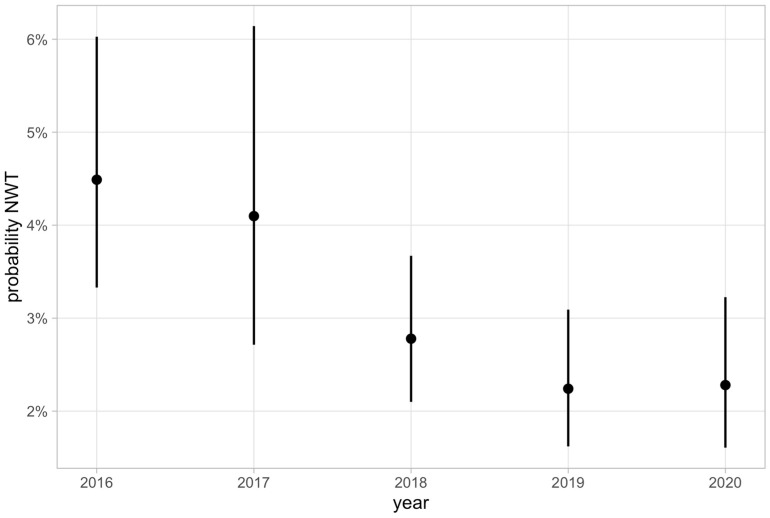
Yearly probabilities and corresponding 95% confidence intervals (CI) for the occurrence of non-wild-type (NWT) isolates calculated for colistin.

**Figure 2 antibiotics-12-01424-f002:**
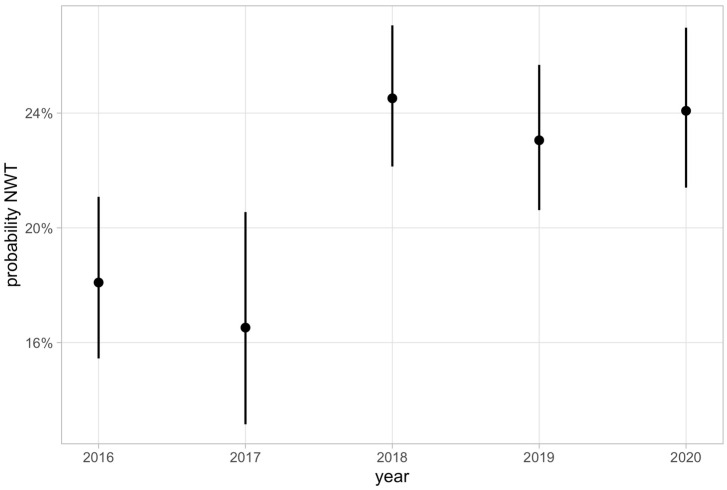
Yearly probabilities and corresponding 95% confidence intervals (CI) for the occurrence of non-wild-type (NWT) isolates for enrofloxacin.

**Figure 3 antibiotics-12-01424-f003:**
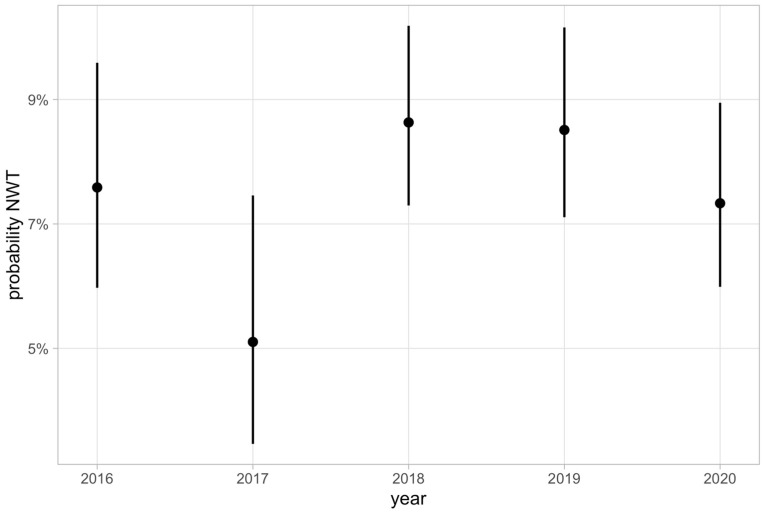
Percentage distribution of ceftiofur mutant and wild-type isolates of *E. coli* between 2016 and 2020.

**Figure 4 antibiotics-12-01424-f004:**
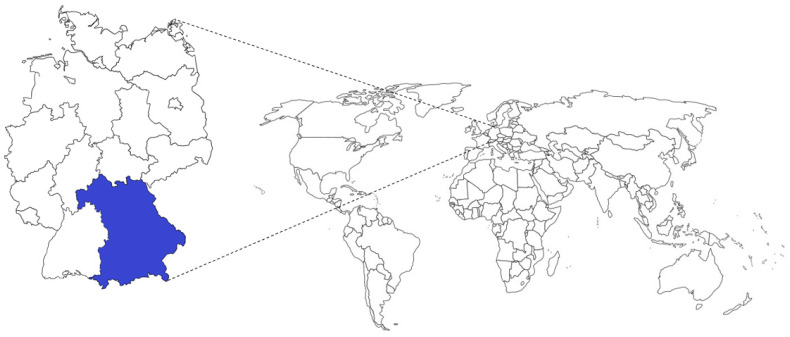
Scheme of Germany (**left**) and the world (**right**) showing the sample origin in Bavaria (shaded in blue).

**Table 1 antibiotics-12-01424-t001:** Observed MIC values (µg/mL) of colistin in *Escherichia coli* isolates between 2016 and 2020. The grey shaded column is representing NWT isolates.

Year		WT Isolates			NWT Isolates
	MIC	<0.5 µg/mL	1 µg/mL	2 µg/mL	>2 µg/mL
2016		83.82%	8.74%	0.80%	6.63%
		(*n* = 834/995)	(*n* = 87/995)	(*n* = 8/995)	(*n* = 66/995)
		*n* = 939/995			*n* = 66/995
2017		78.29%	14.73%	0.78%	6.20%
		(*n* = 404/516)	(*n* = 76/516)	(*n* = 4/516)	(*n* = 32/516)
		*n* = 484/516			*n* = 32/516
2018		81.00%	13.55%	1.55%	3.90%
		(*n* = 1518/1874)	(*n* = 254/1874)	(*n* = 29/1874)	(*n* = 73/1874)
		*n* = 443/516			*n* = 73/516
2019		75.15%	19.97%	1.89%	3.08%
		(*n* = 1291/1718)	(*n* = 343/1718)	(*n* = 31/1718)	(*n* = 53/1718)
		*n* = 1665/1718			*n* = 53/1718
2020		78.89%	17.08%	1.02%	3.01%
		(*n* = 1155/1464)	(*n* = 250/1464)	(*n* = 15/1464)	(*n* = 44/1464)
		*n* = 1420/1464			*n* = 44/1464

**Table 2 antibiotics-12-01424-t002:** Calculated odds ratios, *p*-value and change (95% confidence interval) for the occurrence of colistin non-wild-type isolates (NWT) between 2016 and 2020.

Odds Ratio [95% CI]	*p*-Value	Change [95% CI]
2.01 [1.05–3.87]	0.013	101.4% [4.94–286.52%]

**Table 3 antibiotics-12-01424-t003:** Observed MIC values of enrofloxacin (µg/mL) in *Escherichia coli* isolates between 2016 and 2020. Grey shaded columns are representing NWT isolates.

Year		WT Isolates			NWT Isolates			
	MIC	<0.03125 µg/mL	0.0625 µg/mL	0.125 µg/mL	0.25 µg/mL	0.5 µg/mL	1 µg/mL	>1 µg/mL
2016		74.90%	4.66%	2.33%	3.85%	3.95%	1.42%	8.91%
		(*n* = 740/988)	(*n* = 46/988)	(*n* = 23/988)	(*n* = 38/988)	(*n* = 39/988)	(*n* = 14/988)	(*n* = 88/988)
		*n* = 809/988			*n* = 179/988			
2017		74.46%	6.68%	2.36%	3.34%	4.72%	1.38%	7.07%
		(*n* = 379/509)	(*n* = 34/509)	(*n* = 12/509)	(*n* = 17/509)	(*n* = 24/509)	(*n* = 7/509)	(*n* = 36/509)
		*n* = 425/509			*n* = 84/509			
2018		63.68%	5.07%	2.24%	6.52%	7.75%	2.88%	11.86%
		(*n* = 1192/1872)	(*n* = 95/1872)	(*n* = 42/1872)	(*n* = 122/1872)	(*n* = 145/1872)	(*n* = 54/1872)	(*n* = 222/1872)
		*n* = 1329/1872			*n* = 543/1872			
2019		66.03%	3.85%	2.62%	7.11%	7.40%	1.92%	11.07%
		(*n* = 1133/1716)	(*n* = 66/1716)	(*n* = 45/1716)	(*n* = 122/1716)	(*n* = 127/1716)	(*n* = 33/1716)	(*n* = 190/1716)
		*n* = 1244/1716			*n* = 472/1716			
2020		65.96%	3.49%	2.19%	6.56%	8.61%	2.39%	10.80%
		(*n* = 965/1463)	(*n* = 51/1463)	(*n* = 32/1463)	(*n* = 96/1463)	(*n* = 126/1463)	(*n* = 35/1463)	(*n* = 158/1463)
		*n* = 1048/1463			*n* = 415/1463			

**Table 4 antibiotics-12-01424-t004:** Calculated odds ratios, *p*-value and change (95% confidence interval) for the occurrence of enrofloxacin non-wild-type isolates (NWT) between 2016 and 2020.

Odds Ratio [95% CI]	*p*-Value	Change [95% CI]
0.7 [0.5–0.97]	0.006	−43.56% [−100.75–−2.66%]

**Table 5 antibiotics-12-01424-t005:** Relative observed MIC values (µg/mL) of ceftiofur in *Escherichia coli* isolates between 2016 and 2020. Grey shaded columns are representing NWT (MIC > 1) isolates.

		WT Isolates				NWT Isolates		
	MIC	<0.125 µg/mL	0.25 µg/mL	0.5 µg/mL	1 µg/mL	2 µg/mL	4 µg/mL	>4 µg/mL
2016		3.87%	51.68%	33.98%	1.73%	0.41%	0.31%	8.04%
		(*n* = 38/983)	(*n* = 508/983)	(*n* = 334/983)	(*n* = 17/983)	(*n* = 4/983)	(*n* = 3/983)	(*n* = 79/983)
		*n* = 897/983				*n* = 86/983		
2017		6.07%	52.05%	34.25%	1.76%	0.98%	0.39%	4.50%
		(*n* = 31/511)	(*n* = 266/511)	(*n* = 175/511)	(*n* = 9/511)	(*n* = 5/511)	(*n* = 2/511)	(*n* = 23/511)
		*n* = 481/511				*n* = 30/511		
2018		3.85%	48.77%	33.92%	2.24%	0.85%	0.59%	9.78%
		(*n* = 72/1872)	(*n* = 913/1872)	(*n* = 635/1872)	(*n* = 42/1872)	(*n* = 16/1872)	(*n* = 11/1872)	(*n* = 183/1872)
		*n* = 1662/1872				*n* = 210/1872		
2019		3.15%	51.37%	33.18%	1.46%	0.76%	0.87%	9.21%
		(*n* = 54/1715)	(*n* = 811/1715)	(*n* = 569/1715)	(*n* = 25/1715)	(*n* = 13/1715)	(*n* = 15/1715)	(*n* = 158/1715)
		*n* = 1459/1715				*n* = 186/1715		
2020		2.86%	53.44%	32.26%	1.91%	1.30%	0.34%	7.98%
		(*n* = 42/1466)	(*n* = 782/1466)	(*n* = 473/1466)	(*n* = 28/1466)	(*n* = 19/1466)	(*n* = 5/1466)	(*n* = 117/1466)
		*n* = 1325/1466				*n* = 141/1466		

**Table 6 antibiotics-12-01424-t006:** Calculated odds ratios, *p*-value and change (95% confidence interval) for the occurrence of ceftiofur non-wild-type isolates (NWT) between 2016 and 2020.

Odds Ratio [95% CI]	*p*-Value	Change [95% CI]
1.04 [0.65–1.65]	0.897	3.75% [−53.3–65.03%]

## Data Availability

The data presented in this study are included in the text as well as in Appendix A.

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
