# Peer review of "Antimicrobial Susceptibility from a One Health Perspective Regarding Porcine Escherichia coli from Bavaria, Germany"

_antibiotics, 2023, doi:10.3390/antibiotics12091424_

Round 1
Reviewer 1 Report
General:
It might be advisable to have one figure showing the changes in the quantities of the antimicrobials used per year, and the corresponding resistance rates, in the Discussion section, with respect to (non-)correlation.
line 20: should it read 2016-2020 (five years; cf line 298)
line 23: 3rd+4th
line 54: "it is a widely-..."
line 63: what does "body site" mean?
line 93: I am not sure if it is advisable the present the results before the "Materials and Methods" section
line 93ff / results: the authors study changes from year to year/between years; would the available data allow to study different pig production types, e.g. organic/non-organic; fattening pigs or sows?
line 96: maybe better "were isolated" instead of "originated"
line 259: suggest to delete "as published", since the bracket including 2 references already states that.
line 305: "Species definition.. was determined..." maybe can be written in a simple way: "E. coli were confirmed at species level by fluorescence..."
line 325: "Table 7."
lines 358-359: when excel file submitted together with the manuscript will be available to the readers, the authors should indicate here where this raw data file is accessible.
lines 403-406: article title should not be completely in cap letters
Author Response
Thank you very much for your valuable comments. We have amended the manuscript as followed:
1) It might be advisable to have one figure showing the changes in the quantities of the antimicrobials used per year, and the corresponding resistance rates, in the Discussion section, with respect to (non-)correlation.
ANSWER
The applicant agrees that presenting this non-correlation graphically would be a great benefit for the reader. Unfortunately, the number of antimicrobial sales for each antimicrobial substance group is not shown in the available report (Reference no. 42) but only the amount of reduction over the mentioned period of time was shown. Thus, no year-by-year data are available making it impossible to set up an appropriate figure.
2) line 20: should it read 2016-2020 (five years; cf line 298)
ANSWER: We agreed and corrected this accordingly.
3) line 23: 3rd+4th
ANSWER: We agreed and corrected this accordingly.
4) line 54: "it is a widely-..."
ANSWER: We corrected this accordingly.
5) line 63: what does "body site" mean?
ANSWER: Within the CLSI guidelines, breakpoints are available for each bacterial species, the individual animal species and finally as well according to the organ system that revealed the bacterial infection. The authors of the guidelines name this the “body site” (CLIS, Vet01S, Reference No. 32).
6) line 93: I am not sure if it is advisable the present the results before the "Materials and Methods" section
ANSWER: The authors understand this point- However, we followed the instructions and the template provided by the journal for preparing the manuscript.
7) line 93ff / results: the authors study changes from year to year/between years; would the available data allow to study different pig production types, e.g. organic/non-organic; fattening pigs or sows?
ANSWER: Unfortunately, the herein presented data do not allow to study this area as the data have been handled completely anonymized without any possibility to follow back the production type of the farm. However, we fully agree that this is a very interesting point to be studied and looked at in the future and should be part of future studies. Therefore, we added this as a point to the discussion of our manuscript, please refer to L. 304 – 312.
8) line 96: maybe better "were isolated" instead of "originated"
ANSWER: We agree and changed the wording accordingly.
9) line 259: suggest to delete "as published", since the bracket including 2 references already states that.
ANSWER: We agree and amended the sentence accordingly.
10) line 305: "Species definition.. was determined..." maybe can be written in a simple way: "E. coli were confirmed at species level by fluorescence..."
ANSWER: The authors agree and have amended the section accordingly.
11) line 325: "Table 7."
ANSWER: We corrected this typo accordingly.
12) lines 358-359: when excel file submitted together with the manuscript will be available to the readers, the authors should indicate here where this raw data file is accessible.
ANSWER: Indeed we intended to upload the raw data as an excel file. We will mention this in the end of the manuscript once the upload link will be available.
13) lines 403-406: article title should not be completely in cap letters
ANSWER: We agree and have adapted the title accordingly.
Reviewer 2 Report
Review of Manuscript:
"Antimicrobial Susceptibility in a One Health perspective regarding porcine Escherichia coli from Bavaria, Germany"
Also in the resume you need to indicate the full name and then the abbreviation - Probably should be "Antimicrobial usage (=AMU)" - line 22.
Then, according to the text, it is not necessary to specify the full name again and again the abbreviation - For example "Antimicrobial resistance (AMR)" - line 32 and 36.
The same problem - "ECOFF" - line 74 and 100,
The same - "odd-ratio (OR)" - line 124, 128.
Please check all text!! BUT......All the abbreviations and repetition of abbreviation explanations throughout the text makes it very hard to read. The reader loses the thought, because everything needs to be thought about whether this abbreviation is higher or not. When we write a book, we give a list of abbreviations at the beginning. But here the article is 10 pages of text. Is it really difficult to write the name in full every time and not give abbreviations? I strongly recommend not to give abbreviations, but to write the name in full. Thus, your article will be read with pleasure and scientists will be able to get all the information calmly.
The authors have tried to investigate the problem in detail. The article is very interesting.
Author Response
Thank you very much for your valuable commets. We have amended our manuscript as following:
"Antimicrobial Susceptibility in a One Health perspective regarding porcine Escherichia coli from Bavaria, Germany"
Also in the resume you need to indicate the full name and then the abbreviation - Probably should be "Antimicrobial usage (=AMU)" - line 22.
Then, according to the text, it is not necessary to specify the full name again and again the abbreviation - For example "Antimicrobial resistance (AMR)" - line 32 and 36.
The same problem - "ECOFF" - line 74 and 100,
The same - "odd-ratio (OR)" - line 124, 128.
Please check all text!! BUT......All the abbreviations and repetition of abbreviation explanations throughout the text makes it very hard to read. The reader loses the thought, because everything needs to be thought about whether this abbreviation is higher or not. When we write a book, we give a list of abbreviations at the beginning. But here the article is 10 pages of text. Is it really difficult to write the name in full every time and not give abbreviations? I strongly recommend not to give abbreviations, but to write the name in full. Thus, your article will be read with pleasure and scientists will be able to get all the information calmly.
The authors have tried to investigate the problem in detail. The article is very interesting.
ANSWER:
The authors thank the reviewer for this valid point and tried to improve the readability of the manuscript by excluding abbreviations for the following terms throughout the manuscript: antimicrobial resistance, antimicrobial usage, odd-ratio.
Reviewer 3 Report
The manuscript by Ade and cols. is a concise and quite interesting analysis between the resistance to three different antibiotics and the usage of these antibiotics. Even though the common sense would indicate that the less antibiotic is used the lower the resistance levels are, the authors found that this is not always the case.
Even though the manuscript is fine as it is, it could include information on what other antibiotics were in used at the same time. They mentioned that co-selection exists between fluoroquinolone resistance genes and ß- lactamases resistance genes, among other antibiotics and it may be one of the reasons why the resistance to the fluoroquinolone increased over the last years in spite of the sharp reduction in its usage. By the way, "fluoroquinolones" are mentioned as "fluorochinolons" three times (lines 23, 258 and 317)
On the other hand, it is plausible that longer times will be needed to see a dramatic decrease in resistance. The authors analyzed pretty much the same window of antibiotic usage and antibiotic resistance. However, they may be seeing the resistance that is direct consequence of the antibiotics usage in the previous lustrum or even decade.
Finally, it would be better to show how the antibiotic usage was decreasing year by year and not just the total reduction from 2014 to 2021; perpahs a better correlation might be established.
Author Response
Thank you very much for your valuable comments. We have amended the manuscript as following:
1) The manuscript by Ade and cols. is a concise and quite interesting analysis between the resistance to three different antibiotics and the usage of these antibiotics. Even though the common sense would indicate that the less antibiotic is used the lower the resistance levels are, the authors found that this is not always the case.
Even though the manuscript is fine as it is, it could include information on what other antibiotics were in used at the same time. They mentioned that co-selection exists between fluoroquinolone resistance genes and ß- lactamases resistance genes, among other antibiotics and it may be one of the reasons why the resistance to the fluoroquinolone increased over the last years in spite of the sharp reduction in its usage. By the way, "fluoroquinolones" are mentioned as "fluorochinolons" three times (lines 23, 258 and 317)
ANSWER:
The authors thank the reviewer very much for these definitively valid comments. The authors agree and included information on the fact that penicillins and tetracyclines are the most commonly used antimicrobial substances in pig production into the discussion. Please refer to L. 262 – L. 274. Further, we corrected the spelling of “fluoroquinolones” throughout the manuscript.
2) On the other hand, it is plausible that longer times will be needed to see a dramatic decrease in resistance. The authors analyzed pretty much the same window of antibiotic usage and antibiotic resistance. However, they may be seeing the resistance that is direct consequence of the antibiotics usage in the previous lustrum or even decade.
ANSWER:
The authors agree to this important point. We added this point in the discussion section (L. 271 – 274 and L. 285-288).
3) Finally, it would be better to show how the antibiotic usage was decreasing year by year and not just the total reduction from 2014 to 2021; perpahs a better correlation might be established.
ANSWER:
The authors agree that presenting year-by-year reduction rates would definitively be better than showing just the amount of reduction from 2014 to 2021. However, the available report (Reference No. 42) only shows the amount between the mentioned years and not individual year-by-year numbers. Thus, if data will be available once, this should definitively be a point for extended studies.
Reviewer 4 Report
I would like to congratulate the authors on their study. The aim of this study was to assess the evolution of non-wild type swine E. coli isolates to critically important antibiotics, specifically colistin, enrofloxacin, and ceftiofur, within a defined epidemiological setting over a period of five years.
This constitutes an epidemiological study which, from my standpoint, holds significance for public health as well as scientific knowledge. It is important to note that the authors, contrary to common belief, demonstrate that antibiotic use is not always linked to resistance development.
However, towards the end of the manuscript, the authors highlight the need for including "evaluation of animal transportation routes, as well as biosecurity measures in the farms" to assess antimicrobial resistance. I felt the absence of the authors formulating hypotheses on these aspects, as well as delving a bit deeper into them. At the very least, the main arguments for and against these hypotheses should be highlighted.
Furthermore, in my view, Table 6 appears unnecessary. The information within this table could be incorporated into the text.
Overall, the manuscript is well presented, and the methodology has been aptly conducted. If the queries I have raised are addressed, I would be pleased to accept this manuscript for publication as an article in Antibiotics.
Author Response
Thank you very much for your valuable comments. We have amended the mansucript as following:
1) However, towards the end of the manuscript, the authors highlight the need for including "evaluation of animal transportation routes, as well as biosecurity measures in the farms" to assess antimicrobial resistance. I felt the absence of the authors formulating hypotheses on these aspects, as well as delving a bit deeper into them. At the very least, the main arguments for and against these hypotheses should be highlighted.
ANSWER:
The authors thank the reviewer for this valid point. A better explanation concerning the raised issue was included into the discussion of the manuscript, please refer to L. 304 - 312.
2) Furthermore, in my view, Table 6 appears unnecessary. The information within this table could be incorporated into the text.
ANSWER
The authors suggest that the question raised by the reviewer concerns Table 7 instead of Table 6 (as Table 7 indeed only includes very few information while Table 6 is equally to Tables 4 and 2). Therefore, we replaced Table 7 and incorporated the content into the text of the manuscript.